# Undergoing lignin-coated seeds to cold plasma to enhance the growth of wheat seedlings and obtain future outcome under stressed ecosystems

Abd Elfattah T. Elgendy[1], Hesham Elsaid[1], Hani S. Saudy[2]*, Nimer Wehbe[3],
Mohamed Ben Hassine[3], Ruba Al-Nemi[4], Mariusz Jaremko[5]*, Abdul-Hamid Emwas[3]

**1** Faculty of Science, Physics Department, Ain Shams University Cairo, Cairo, Egypt, **2** Faculty of Agriculture, Agronomy Department, Ain Shams University Cairo, Cairo, Egypt, **3** Core Labs, King Abdullah University of Science and Technology (KAUST), Thuwal, Kingdom of Saudi Arabia, **4** Biological and Environmental Science and Engineering (BESE) Division, Bioscience Program, King Abdullah University of Science and Technology (KAUST), Thuwal, Kingdom of Saudi Arabia, **5** Division of Biological and Environmental Sciences and Engineering (BESE), Smart-Health Initiative (SHI) and Red Sea Research Center (RSRC), King Abdullah University of Science and Technology (KAUST), Thuwal, Jeddah, Saudi Arabia

* hani_saudy@agr.asu.edu.eg (HSS); mariusz.jaremko@kaust.edu.sa (MJ)

## Abstract

Climate changes threat global food security and food production. Soil salinization is one of the major issues of changing climate, causing adverse impacts on agricultural crops. Germination and seedlings establishment are damaged under these conditions, so seeds must be safeguard before planting. Here, we use recycled organic tree waste combined with cold (low-pressure) plasma treatment as grain coating to improve the ability of wheat seed cultivars (Misr-1 and Gemmeza-11) to survive, germinate and produce healthy seedlings. The seeds were coated with biofilms of lignin and hash carbon to form a protective extracellular polymeric matrix and then exposed them to low-pressure plasma for different periods of time. The effectiveness of the coating and plasma was evaluated by characterizing the physical and surface properties of coated seeds using X-ray photoelectron spectroscopy (XPS), scanning electron microscopy (SEM), nuclear magnetic resonance (NMR) spectroscopy, and wettability testing. We also evaluated biological and physiological properties of coated seeds and plants they produced by studying germination and seedling vigor, as well as by characterizing fitness parameters of the plants derived from the seeds. The analysis revealed the optimal plasma exposure time to enhance germination and seedling growth. Taken together, our study suggests that combining the use of recycled organic tree waste and cold plasma may represent a viable strategy for improving crop seedlings performance, hence encouraging plants cultivation in stressed ecosystems.

**Data Availability Statement:** All relevant data are within the manuscript and its Supporting Information files.

**Funding:** The research was funded via Science & Technology Development Fund (STDF), Ministry of Scientific Research Egypt, for financial support and providing required research facilities through the approved project; ID: IG-43568.

**Competing interests:** The authors have declared that no competing interests exist.

# 1 Introduction

Agricultural wastes contribute to an increase in greenhouse gases, especially $CO_2$, which is responsible for global warming. It has been documented that using the raw materials of plant residues as soil amendments caused increase in $CO_2$ and $N_2O$ fluxes [1,2]. There is a positive correlation between the amount of plant raw materials and gas emissions, sharing in climate change issues [3]. The agricultural wastes are regarded as one of the remarkable sources of greenhouse gas emissions that should not be disregarded [4]. For shrinking their undesirable impacts, wastes can be recycled to obtain beneficial by-products used in agriculture [5–7]. The obtainable by-products are exploited in several agronomic purposes [8–10]. In this respect, using these by-products, instead of raw wastes, showed reductions in gas emissions amounted to 70% [11–13]. Lignocellulose is the most abundant renewable raw material, but only about 4.4% of it is exploited annually [14]. Lignin, a significant component in lignocellulosic materials, is an amorphous heteropolymer contains phenylpropane units linked by diversified bonds [15,16]. Despite there are several researches pointed out the beneficial uses and recycling of lignocellulosic materials, knowledge about the importance of lignin as a seed coating is unavailable

On the other hand, the stressed environments are considered the serious challenge for agriculture in arid regions with limited access to freshwater. These regions commonly encounter harmful ecological issues such as drought, extreme heat, nutrient shortages, and soil salinization [17–23]. Such stresses correlate with disturbance in plant physiology, hence poor crop growth, development and yields [24–28] and are expected to have a greater effect with global warming, risking food shortages [29,30]. However, salinity is the most dangerous stressor for crop growth, since high salts in soil could cause toxicity, in addition to physiological drought [31–34].

Three main cereal crop species, wheat (*Triticum aestivum*), rice (*Oryza sativa*), and maize (*Zea mays*) are widely grown around the world and feed more than 50% of the global population. However, present crop breeding technologies and artificial selection are insufficient to tackle the booming worldwide population and deteriorating arable farmland. To solve this issue, new strategies, such as novel smart agriculture approaches, are required to enhance the ability of crop plants to adapt to harsh environments.

One such emerging strategy is non-thermal plasma (NTP) or cold plasma treatment under low temperature and pressure [35–39]. NTP treatment is a novel class of ecologically benign methods for the secure and efficient surface decontamination and protection of crop seeds. Numerous studies have already investigated the impact of NTP on the chemistry and morphology of the seed, as well as on DNA, plant resilience to drought, and metal toxicity [40], suggesting that NTP treatment improves seed properties. Thus, plasma-treated seeds are viewed as a promising option for farming in challenging environments and soils with abnormal salinity [41–45]. However, as NTP generates a wide range of positively and negatively charged particles, and reactive oxygen and nitrogen species that may have side effects to seeds on the molecular (DNA) level, care needs to be taken to optimize plasma exposure times. Furthermore, shorter exposure times to plasma may not be sufficient to produce beneficial effects. Therefore, one of the remaining challenges in the field is the need to optimize the plasma treatment time to minimize potential unwanted effects of the NTP and decrease the costs, while maintaining the beneficial effects of seed treatment.

Based on the unique properties of lignin and plasma, the current work hypothesized that undergoing lignin-coated seeds to cold plasma could enhance the seed germination and growth of wheat seedlings grown in salt-affected soil. To test this hypothesis; this study aimed to examined how treatment with different doses of NTP affects the lignocellulosic biomaterial

coating thicknesses around wheat seeds, and how the coating thickness affects the seeds. Furthermore, after plasma treatment, seed germination and seedling growth and physiology under saline soil condition were measured.

## 2. Materials and methods

### 2.1 Materials

**2.1.1 Seed treatment.** In every trial, seeds of two different wheat cultivars (Cultivar 1: Misr-1 and Cultivar 2: Gemmeza-11) were used. As a control (D(Control)), the control seeds were not treated with chemical lignocellulosic biomass materials or plasma. The lignocellulosic biomass material treatment was done. Viscous lignin and hash carbon are mixed, and then wheat seeds are added. This material is gently mixed to create a homogeneous surface around the seeds to obtain lignin-coated seeds. To protect the seeds from the wet environment, the lignin material containing hash carbon needs to be viscous and non-liquid. The seeds were left in the sun for a day to allow the surface-treatment material to dry. We start preparing the seeds for plasma exposure at different times. After coating seeds with biofilms of lignin and hash carbon to form a protective extracellular polymeric matrix, they were exposed to low-pressure plasma called also non-thermal plasm (NTP). NTP treatments were done over one of four time periods: one minute (D(00)), two minutes (D(22)), three minutes (D(33)), or four minutes (D(44)), as described below in 2.1.3.

**2.1.2 Chemical lignocellulosic biomass materials.** Lignocellulosic biomass in crushed form (roughly ~4 mm in size) was purchased from a mill and then sieved to produce a powder with particle sizes of 250–500 μm. Flavone ligninocellulose (FLC) was prepared by a one-pot reaction method [46–48]. Co-monomers including acrylamide and vinyl from treated–grafted polymer were purchased from Sigma-Aldrich as a compatibilizer [49]. All other chemicals were obtained from commercial sources and were of reagent grade. For the nano-fibrillation of FLC powder, 10 g of untreated LC powder was soaked in distilled water for 48h. The suspension of LC powder (1 wt%) was then agitated using a high-speed blender (Vitamix TNC5200, Vita-Mix Corporation, Cleveland, OH, USA) equipped with a 2 L SUS container (X-TREME CAC90B, 143 WARING, East Windsor, NJ, USA) at a stirring speed of 37,000 rpm. After stirring for 45 min, nanofibrillated fiber was obtained by centrifugation at $8,000 \times g$ for 20 min at 25˚C. The recovered wet fiber was infiltrated with t-butyl alcohol and freeze-dried [46,49,50].

For the pretreatment of FLC powder, 10 g of ligninocellulose (LC) was mixed with 10 g of 3.4-dihydroxyacetophenone and 10 mL dimethyl sulfoxide (DMSO as cosolvent) and heated in a dry oven at 110˚C for 16 h. The reaction mixture was then suspended in 100 mL of deionized water and centrifuged at $8,000 \times g$ for 20 min at 25˚C. This washing step was conducted five times to remove the 3.4-dihydroxyacetophenone/DMSO solution. The product was dried in an oven at 90˚C for 24 h and grounded into a powder using a mill.

For alkylation, 5 g of bagasse sample was mixed with 10 g of chloroacetic acid in a 100-mL pressure-proof autoclave reactor (microwave synthesis and extraction, MW-US-UV syn, Co. Ltd., Nagoya, Japan). The reactor was heated in a rotary oven at 140˚C. After heating for 3 h, the modified FLC was washed three times with acetone and then twice with distilled water. To remove unreacted chloroacetic acid, FLC acetic acid was centrifuged at $13,000 \times g$ for 20 min at 25˚C followed by removal of the supernatant. To prepare the composites, a mixture of 18.8 g of thiosemicarbazide and 1.2 g of FLC acetic acid in 500 mL of xylene was stirred at 130˚C for 1 h to obtain a homogeneous solution. The suspension was cast onto a tray covered with polyamide film. After the evaporation of xylene at room temperature in a well-ventilated hood, the resultant dry mixture (thiadiazole biofilm of FLC sample = 47:3:50) was cut into pieces and compounded using a co-rotating twin-screw micro extruder with a recirculating channel

(MC5, Xplore Instruments, Sittard, Netherlands) for 3 min at 180°C and 60 rpm. The extruded strands were cooled in air and pelletized. The pellets were injection-molded into dumbbell-shaped specimens (JIS K7161) (Japanese Standards Association, 2014). The injection and mold temperatures were 190°C and 100°C, respectively [47,48,50,51].

**2.1.3 Using plasma.**   An alternating current (AC) plasma system was used to coat the seeds. The device generates reactive ion species in gases such as argon, helium, oxygen, and nitrogen at low temperature under a low vacuum (25–90 Pa). Air gas can produce different reactive species like nitrogen, oxygen, carbon and OH groups to enhance the surface of the coated seeds and protect it [35,39]. These species react with the seed's surface to generate different chemical entities that can act as a fertilizer and protectant, including making the seeds more tolerant to salinity. In addition, using air gas under low vacuum is more chemically effective and considerably lowers the cost when considering larger scale production. The physical parameters of the AC plasma discharge were pressure $(p) = 25 - 90$ Pascal, $T_G = 300\ K$, applied voltage $(V(t)) = 400$ Volt, and ion current flux $(e\psi_i) = \ldots0.5..\text{mAm}^{-2}$, $T_e = 3\ eV$. Power (P) was set to 41–50 W.

The ion flux and ion energy of the reactive species, such as nitrogen, oxygen, and helium, can be detected. The ion flux and energy (Eqs 1 and 2) were calculated as follow:

$$\text{Ion flux}: \qquad \Gamma_i = 0.61 n_0 \sqrt{T_e/m_i} \tag{1}$$

$$\text{Ion energy}: \quad \varepsilon_i = \frac{eT_e}{2}\ln\left(M/(2.3m_e)\right) \tag{2}$$

where $n_e$ is plasma density, $T_e$ is thermal energy, and M is the ion mass of the reactive species. The ion energy of reactive air species in our study such as nitrogen, and oxygen work at lower energy (2.1–5 eV) which may be more effective for soft biological treatment and not harmful for our seed treatments.

## 2.2 Methods

**2.2.1 Wettability measurement.**   For wettability measurements, 1 µL water droplets were placed on the wheat grain surfaces, and bright field images of all treated and untreated grains in the same frame were taken using a standard stereomicroscope. The diffuse area of the droplets on the surface indicates wettability (Eq 3)

$$\gamma_S = \gamma_L.\cos\theta + \gamma_{SL}. \tag{3}$$

Where: $\gamma_S$: is the solid surface tension, $\gamma_L$: is the liquid surface tension, $\gamma_{SL}$: A solid -land boundary surface tension:

**2.2.2 Scanning electron microscopy (SEM).**   The morphology of the seeds was studied using a Quattro ThermoFisher SEM equipped with a field emission gun (FEG). SEM images were recorded at an accelerating voltage of 5 kV using a secondary electron detector after sputter coating the seeds.

**2.2.3 X-ray photoelectron spectroscopy (XPS).**   To determine the elemental composition and the chemical state of the seeds, an XPS system (Kratos Axis Supra equipped with a monochromatic Al Kα X-ray source (hν = 1486.6 eV) operating at a power of 75 W and under UHV conditions in the range of $\sim 10-9$ mbar was employed. All spectra were recorded in hybrid mode using electrostatic and magnetic lenses and an aperture slot of 300 µm × 700 µm. The wide and high-resolution spectra were acquired at fixed analyzer pass energies of 80 eV and 20 eV, respectively. The samples were mounted in floating mode to avoid differential charging.

**2.2.4 Nuclear magnetic resonance (NMR) spectroscopy.**   All NMR spectra were recorded using a Bruker 400 MHz AVANACIII NMR spectrometer equipped with 4 mm Bruker MAS

probe [52]. The seeds were ground to fine powder using an electrical grinder, and then the grain powder was packed in to 4mm zirconium oxide MAS rotor. To create comparable data all experiments were recorded under the same conditions using the same instrumental parameters described previously [53,54]. The data acquisition and analyses were performed using Topspin 3.5pl7 software (Bruker BioSpin, Rheinstetten, Germany).

**2.2.5 Germination and seedling growth test.** The germination test was conducted using 4 replications of 100 seeds as prescribe by International Seed Testing Association (ISTA). Using top-of-paper germination method, two tested cultivars (Misr-1 and Gemmeza-11) were placed on 15 cm diameter petri lined with filter paper (#1 Whatman International, Maidstone, UK) as the media. The filter paper was initially moistened with 7 mL of distilled water, and an additional 5 mL of water was added on the fifth and ninth days of the experiment. Four plasma exposure times were applied plus the control (D(control), D(00), D(22), D(33), and D(44)). For each cultivar, the plasma treatments were arranged in a completely randomized design with four replicates in ambient laboratory conditions (20±1˚C). Full germination was obtained for all tested treatments (100%). After15 days, the seedlings were isolated to measure the radical length, plumule length and seedling length. Seedling dry weight was recorded after oven drying at 105˚C for 24h.

**2.2.6 Greenhouse trial and physiological studies.** At the Faculty of Agriculture, Ain Shams University, the plasma-treated wheat seeds were planted in pots using the randomized complete block design in four replicates. On 2 December 2021, five plasma-treated seeds of wheat cultivars Misr-1 and Gemmeza-11 were planted in plastic pots of 30-cm diameter filling with 6 kg saline soil (salinity level was 5.5 dS m$^{-1}$). At 60 days of age, plant leaf samples were taken to estimate chlorophyll a (ch-a), chlorophyll (ch-b), total chlorophyll, and carotenoids [55] as well as malondialdehyde, MDA [56] and hydrogen peroxide, $H_2O_2$ [57].

## 2.3 Statistical analysis

Using Costat software, version 6.303 (CoHort Software, Monterey, CA) the analysis of variance (ANOVA) for obtained data was performed [58], Based on the randomized complete block design in four replicates, plasma treatment and wheat cultivar were considered as fixed effects, and replications (blocks) were considered as random effects. Mean separation was performed only when the F–test showed significant (P≤0.05) differences among the treatments based on Duncan's multiple range test.

## 3 Results

### 3.1 Non-thermal plasma composition

The spectroscopic analysis revealed that higher voltage levels generated higher levels of ionization, resulting in higher emission intensity (Fig 1, left panel). Furthermore, it has been observed that higher voltage led to a higher concentration of reactive species such as nitrogen, oxygen, and iron in our plasma (Fig 1, right panel). Based on these results we conducted all our NTP treatment experiments using 400 V, as these conditions corresponded to suitable reactive species that could interact safely with seeds

### 3.2 Effect of plasma dose on wheat seedling growth

First, it should be mentioned that plasma did not show any harmful impacts on germination percentage of wheat seeds, as complete emergence was obtained under all tested plasma doses for both wheat cultivars. Generally, findings revealed that 2 minute NTP treatment produced optimal results for both tested cultivars (Table 1). It has been noted that 1 minute treatment

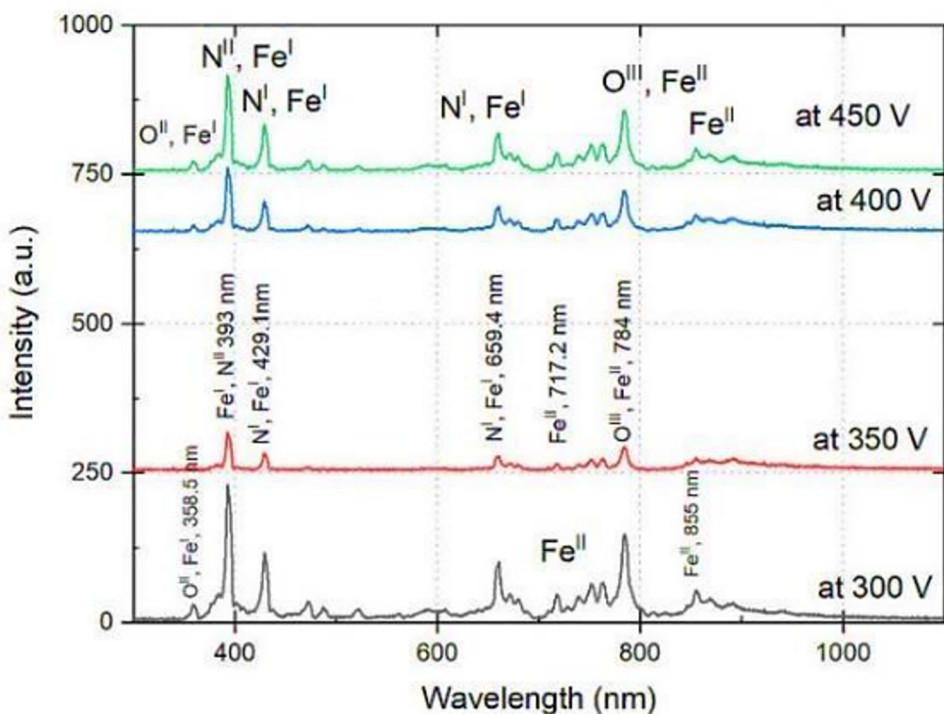

**Fig 1. Spectroscopic analysis of AC plasma of air during coating at different applied voltages.** (Left) The panel shows the effect of the applied voltage on plasma intensity and composition. As expected, the intensity and diversity of species generated increased with increased voltage, which is also seen in the panel on the right.

was not sufficient to reach the maximum beneficial effect (seen at 2 minutes); however, even the 1 minute treatment resulted in some improvement over "no treatment" conditions. Additionally, both 3 and 4 minute treatments led to some deterioration of seedling growth, when compared to the 2 minute dose. Results also illustrated some possible differences between the two cultivars, since Misr-11 exhibited differences between the 3- and 4-minute treatments, while Gemmeza-11 showed no differences between the 3- and 4-minute treatments. These results suggest that conclusions based on one cultivar may not translate to others, and that

**Table 1. Seedling growth rate of wheat cultivars (Misr-1 and Gemmeza-11) results at different plasma exposure times and its correlation coefficient.**

| Variable | Plasma treatment (time / min) | Seedling growth rate | Correlation Coefficient | Contact area on seed surface (A/ $\mu m^2$) |
|---|---|---|---|---|
| Misr-1 cultivar | | | | |
| | No Treatment | Normal | 0.2–0.4 | 101837 |
| | 1 min | Improved | 0.6–0.75 | 113596 |
| | 2 min | Better improvement | 0.75–1.00 | 68762 |
| | 3 min | Improved | 0.6–0.75 | 89745 |
| | 4 min | Progress | 0.4–0.6 | 137546 |
| Gemmeza-11 cultivar | | | | |
| | No Treatment | Normal | 0.2–0.4 | 83456 |
| | 1 min | Improved | 0.6–0.75 | 76892 |
| | 2 min | Better improvement | 0.75–1.00 | 112564 |
| | 3 min | Improved | 0.4–0.6 | 84535 |
| | 4 min | Progress | 0.4–0.6 | 141002 |

each cultivar needs to be individually evaluated to determine the most optimal NTP treatment conditions.

For Misr-1, just the treatment with lignocellulosic biomass (D(00)) was sufficient to result in increases in radical length, plumule length, seedling length and seedling dry weight of 7.9%, 18.1%, 12.0% and 19.7%, respectively (Fig 2 shows just the effects on seedling length and dry weight; results regarding radical length and plumule length are shown in Supplemental Information, as is the Table 1 with exact values from which we derived the % values mentioned in

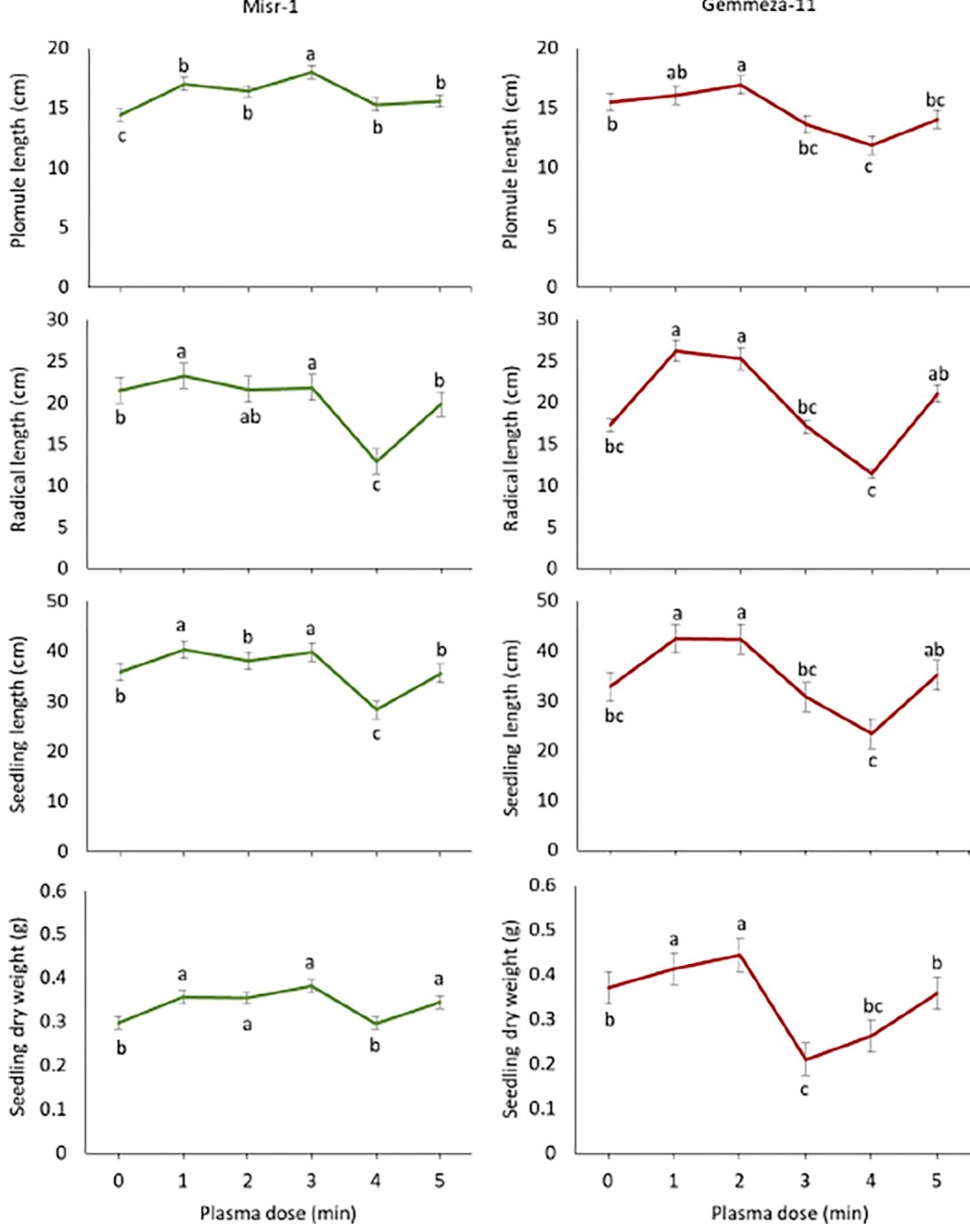

**Fig 2.** Seedling growth of Misr-1 (A) and Gemmeza-11 (B) wheat cultivars as affected by plasma dose treatments measured after one month of growth in a petri dish Values are the mean of 3 replicates ± standard errors. Points with different letters are statistically significant at p ≤ 0.05.

the text). In these experiments with Misr-1, the best increases under NTP dosing conditions were obtained with D(33) (1.2%, 25.0%, 10.9% and 28.1%). For Gemmeza-11 (Fig 2), we observed more dramatic effects on seedling length, as D(00) increased the values by 51.4%, 3.9%, 29.3% and 12.0%, respectively, and D(22) increased them to 46.2%, 9.0%, 29.0% and 19.7%. For both cultivars, D(44) was detrimental for their growth, and, additionally, Gemmeza-11 appeared to be especially sensitive to optimal treatment. Taken together, these studies showed that combined treatment of wheat seeds with recycled tree waste (lignocellulosic biomass) and low-pressure plasma (NTP) has the potential to improve seedling growth of wheat.

### 3.3 Wettability

To identify potential reasons for the improvements which observed in our seedling studies, the seed surface wettability, a very important parameter for germination, was assessed. Thus, improving and increasing the seed surface wettability is a desirable step in seed treatment approaches. In this context wettability revealed that the contact areas of 1 µL size water droplets with seeds treated at different NTP exposure times are shown for Misr-1 in and for Gemmeza-11 in Fig 3 (see also Table 1). Misr-1 seeds treated with a 2 minute NTP dose (D(22)) had reduced wettability compared to D(control) and D(00). In contrast, D(33) or D(44) treated samples had increased levels of surface wettability. On the other hand, Gemmeza-11 seeds had fluctuating wettability levels. The more fluctuating results of the Gemmeza-11 show that its higher sensitivity to plasma treatment when compared to Misr-1, which is aligned with what we observed above in seedling growth experiment.

To quantify the seed wettability, we calculated the values of water droplet contact areas on the surface of controlled and plasma treated seeds Misr-1 and Gemmeza-11 (Table 1). In agreement with microscopy based observations, we noted that the water/seed contact area was significantly decreased in Misr-1 under D(22) dose conditions, and that effects in Gemmeza-11 are much more variable across the NTP treatment conditions. From these observations, it can be concluded that the exposure time affects wettability and nutrient accumulation. Notably, D(00), D(22), and D(33) had the strongest effect on both Misr-1 and Gemmeza-11 cultivars. However, correlating wettability with other seedling traits, including radical length, plumule length, seedling length, and seedling dry weight was less clear.

### 3.4 Solid state nuclear magnetic resonance (NMR) characterization of treated seeds

For examining whether combined treatment of lignocellulosic biomass and NTP led to specific structural and chemical changes on the seed, NMR spectroscopy was applied. Given that solid state NMR has not previously been used for wheat seed analysis, we had to develop a sample preparation procedure that would retain seed integrity and allow for accurate acquisition of the NMR data. As seen in supplementary materials (S1 Fig), the seed samples yielded well-disperse solid-state NMR spectra, clearly showing the distribution of the signals corresponding to chemical nature of carbon (aliphatic region between 0 and 100 ppm; aromatic region above 150 ppm). We noticed that the one minute treated sample in red (S1 Fig) displayed a slight increase in NMR peak intensity in aliphatic region where signals from alkyl groups can be found (0–50 ppm) and in the aromatic region (around 175 ppm) when compared to the control (in blue; S1 Fig).

### 3.5 SEM analysis of Misr-1 and Gemmeza-11 cultivar seeds

To investigate whether the treatment changed the structure of the seed's coat in a manner that may explain improvements in seedling growth we used scanning electron microscopy (SEM).

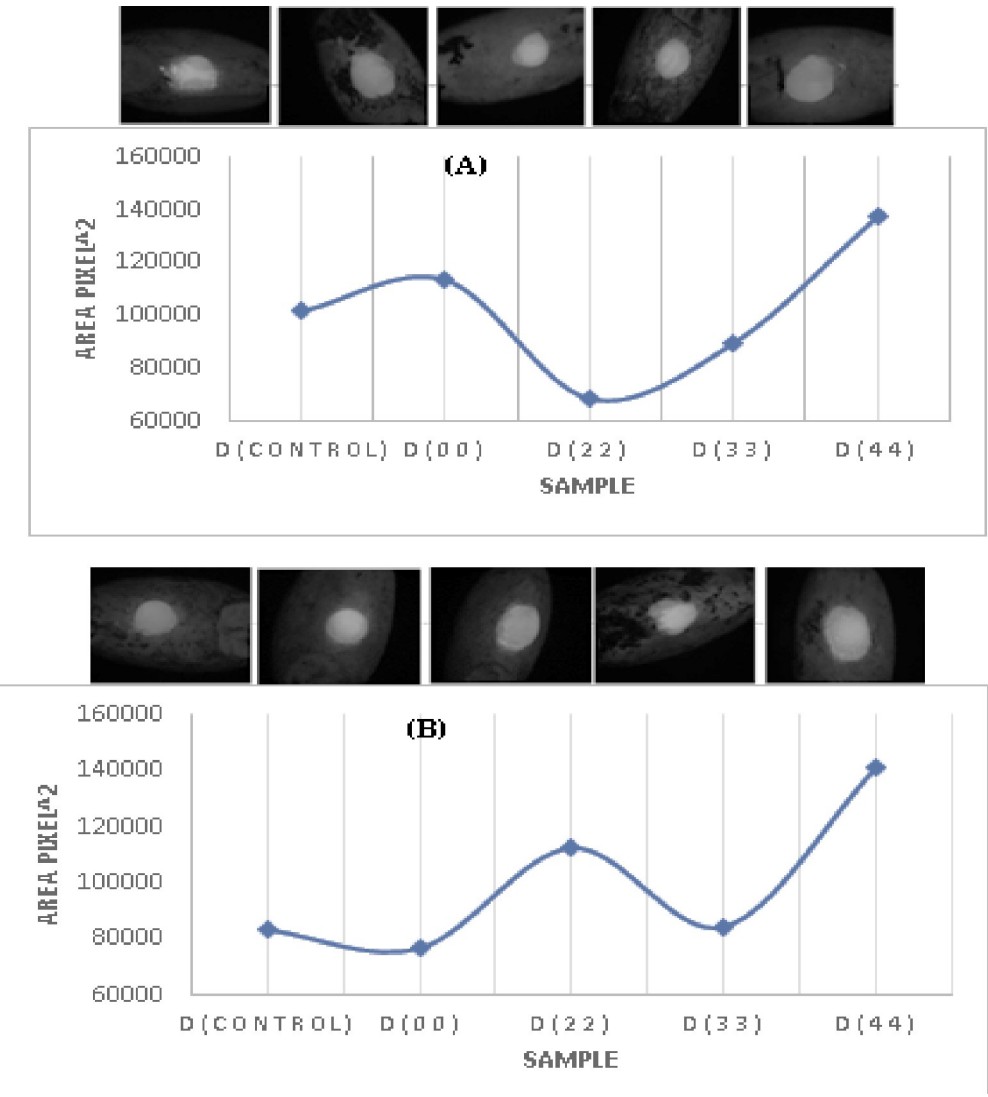

**Fig 3.** Relationship between the contact area of water droplets on the surface of Misr-1 (A) and Gemmeza-11 (B) wheat seed (area) and different NTP doses (1 min: D(control), 2 min: D(00), 3 min: D(22), 4 min: D(33), 5 min: D (44)).

This technique allowed visualizing the finer structural detail of the wheat seed surface, and comparing the features of the two cultivars under different treatment conditions. Fig 4A–4A shows SEM images of the Misr-1 samples, including the untreated seed surface at different scales. For D(00), a very thin coating is observed at the top of the seed with 1–10 μm particles (arrows in Fig 4B–4B"). For D(22), a coating of about 30-μm thickness covers almost the whole seed (arrows in Fig 4C–4C"). D(33) resulted in a thin coating covering a few hundred micrometers of the seed surface (Fig 4D–4D"). For D(44), the coating is barely seen, and only some small particles of a few micrometers in size are observed at the top surface (Fig 4E–4E"). Therefore, we conclude that for Misr-1, D(22) conditions produced the most extensive coating, which may explain the superior seedling growth rate that we noted above. Fig 5 shows SEM images of Gemmeza-11 samples. For D(00) and D(44), a significant coating of about 50 μm was identified at the top surface of the seeds (arrows in b-b" and e-e"). For D(22) and D

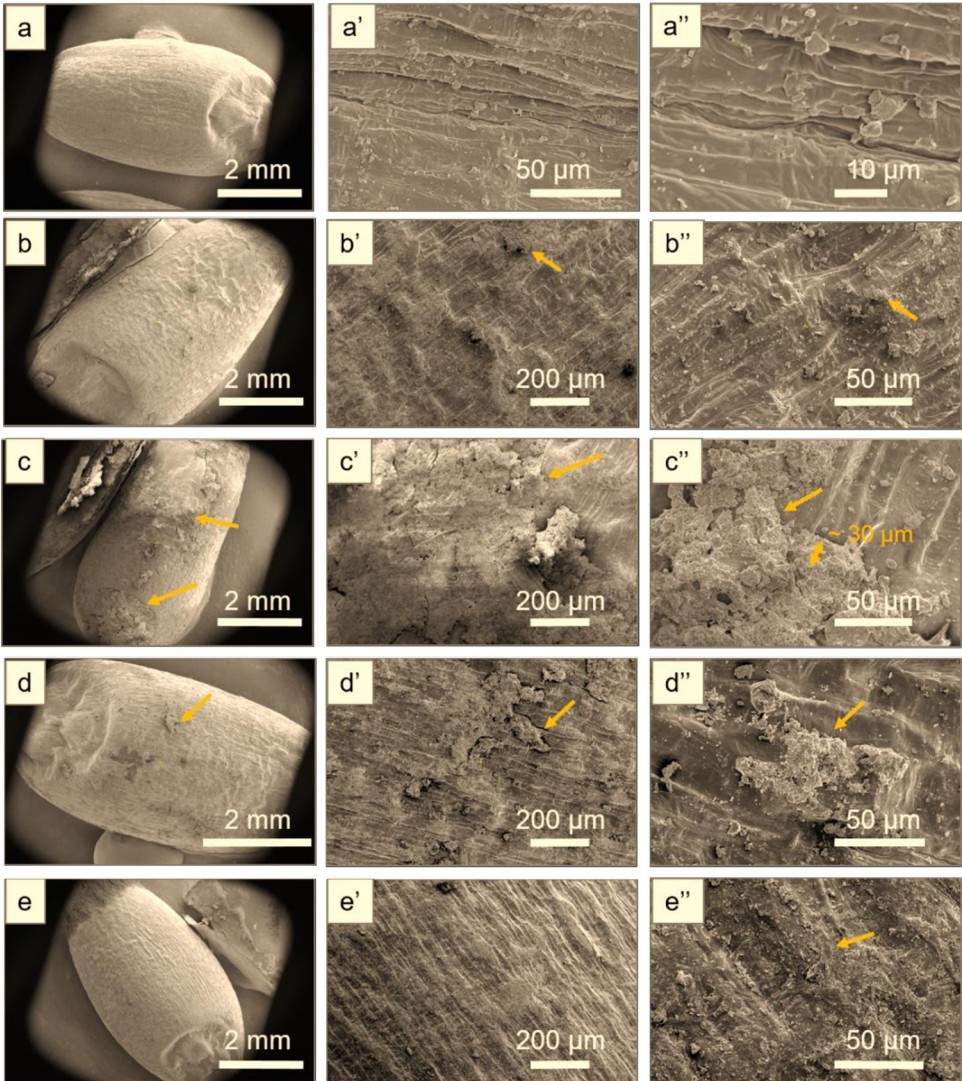

**Fig 4.** SEM images of Misr samples: **(a-a")** D control, **(b-b")** D(00), **(c-c")** D(22), **(d-d")** D(33), and **(e-e")** D(44). The images are shown at different scale as indicated.

(33), the coating was very thin, and only a few particles of about 1 μm in size were present (Fig 5A–5A to 5E–5E" and 5D–5D"). Therefore, for Gemmeza-11, the link between coating and the results of the seedling growth tests do not seem to be clear at this point. Moreover, these results further highlight the cultivar-dependent response to the coating process.

### 3.6 XPS analysis of Misr-1 and Gemmeza-11 cultivar seeds

Lastly, to gain deeper insights into how the treatment changed the chemical composition of the seeds' surface, X-ray photoelectron spectroscopy (XPS) was employed. XPS is a technique that allows elemental analysis of surfaces with high sensitivity. XPS spetra of Gemmeza-11 samples indicated that all seeds have a very similar composition independent of NTP exposure time (Fig 6). However, compared to the control, samples exposed to NTP showed a slight decrease in carbon content and an increase in oxygen content. Also, there was a significant decrease in N content for D00 with respect to other samples. In terms of the elemental content,

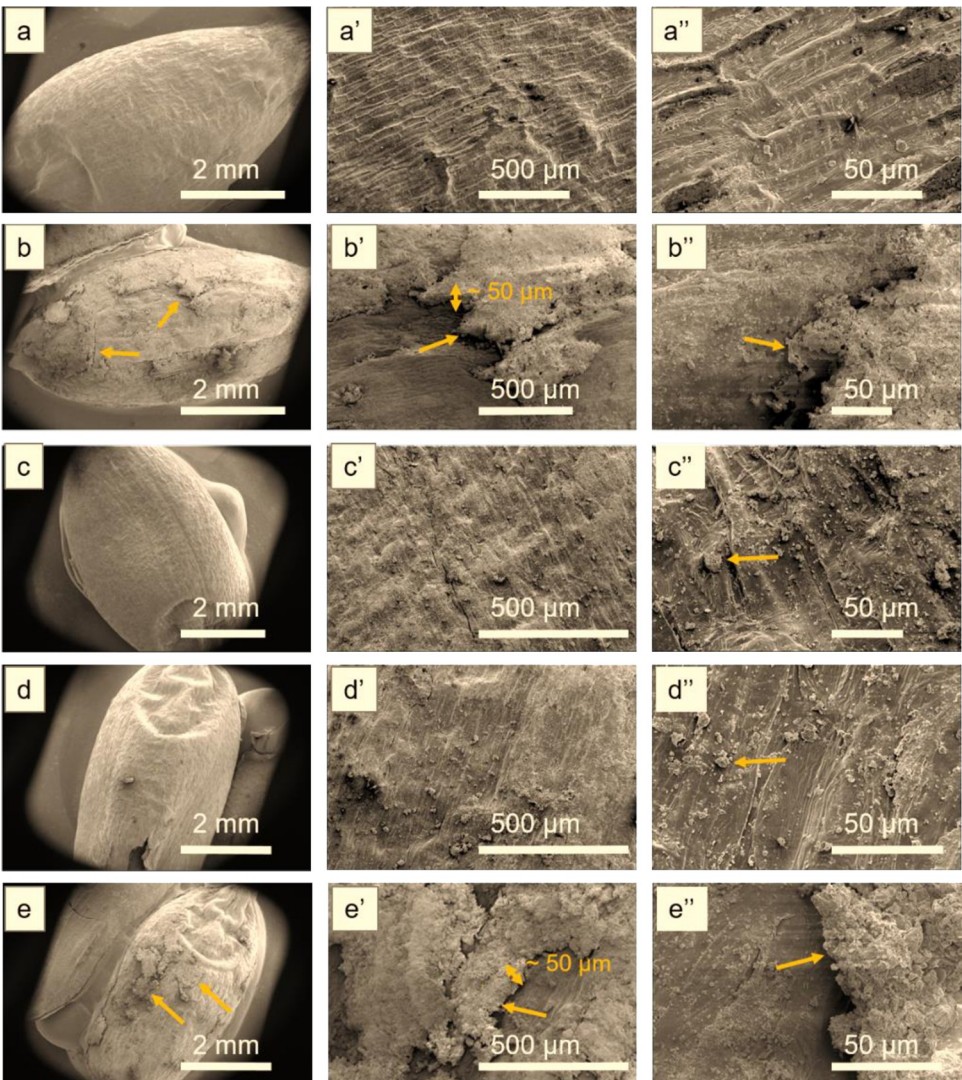

**Fig 5.** SEM images of Gemmeza-11 samples: **(a-a")** D control, **(b-b")** D(00), **(c-c")** D(22), **(d-d")** D(33), and **(e-e")** D(44). The images are shown at different scale as indicated.

C, N, and O elements account for about 95–98% of the composition of each sample irrespective of NTP exposure time, and for 99% of the untreated sample. NTP led to the inclusion of several elements on the wheat surface, but the overall concentration of these elements remained relatively low. The chemical structure of Gemmeza-11 samples did not undergo significant changes following NTP exposure according to the C 1s high-resolution spectra (See supplementary materials Figs 2 and 3), who's peak shape was very similar for all samples. The main carbon bonds assigned to C-N, C-O (detected at around 286 eV), C = O and O-C = O (detected at around 288 eV) exhibited similar shapes and intensities independent of the NTP exposure.

In comparison, Misr-1 showed some different trends (Fig 7). Again, the main composition of NTP-exposed samples was more than 97% C, N and O, and 100% for the control seed indicating potential variability in sed surface composition between cultivars. However, there was some variations in the C/O ratio after exposure to NTP. Misr-1D(00) had the highest C/O

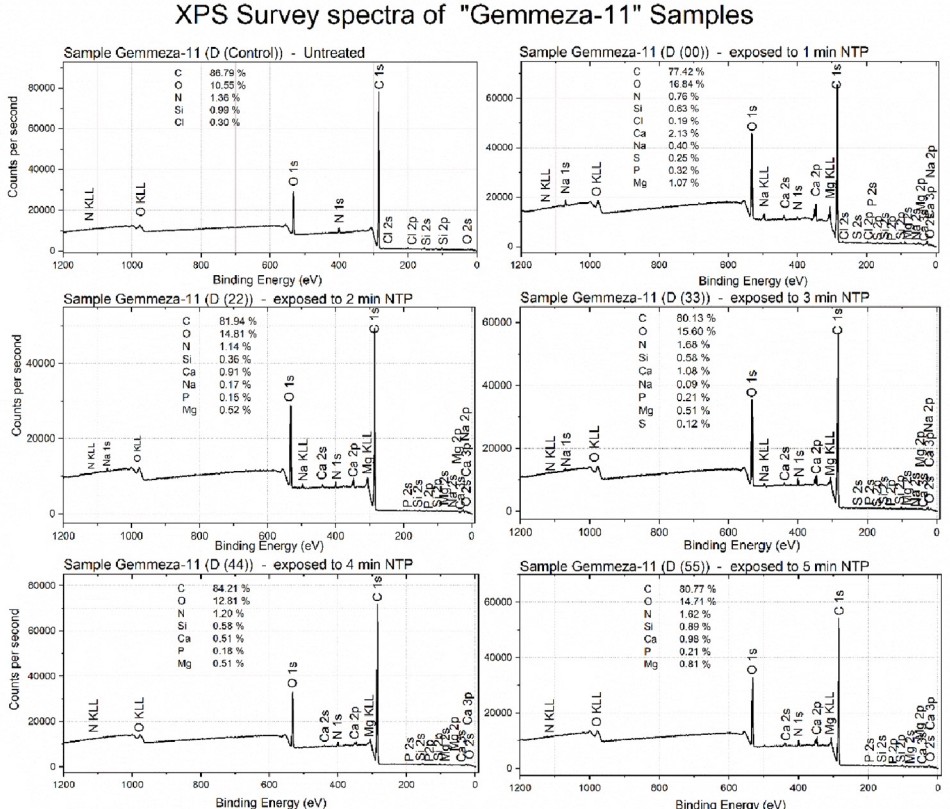

**Fig 6. XPS spectra of Gemmeza-11 seed samples treated under different conditions (D(00), D(11), D(22), D(33) and D(44), as well as the control sample.**

ratio (and thus the lowest oxygen content), whereas Misr-1D(22) had the lowest. The spectra also confirmed the insertion of several elements at low concentration to the wheat surface following NTP exposure. In contrast to Gemmeza-11, where the chemical structure was very similar for all samples irrespective of the exposure, the Misr-1 samples showed clear dose-dependency (Fig 7). The main chemical bonds assigned to C-N, C-O (detected at around 286 eV), C = O and O-C = O (detected at around 288 eV) were less pronounced for samples D(00) and D(44) compared to other doses. This observation is consistent with the lower O and N content of both samples, as shown by the atomic percentages shown in the XPS spectra (Fig 7).

## 3.7 Greenhouse trial and physiological studies

Based on the above observations, chlorophyll a (ch-a), chlorophyll (ch-b), total chlorophyll, carotenoids, malondialdehyde (MDA) and hydrogen peroxide ($H_2O_2$) levels of Misr-1 and Gemmeza-11 seedlings produced from three plasma doses (D00, D22 and D33) were estimated (Table 2). It is worthy to note that NTP treatment showed positive effects on photosynthetic pigments (ch-a, chlorophyll, total chlorophyll, and carotenoids). For Misr-1 cultivar seedlings D(00)-treated showed the most ch-b, D(22) showed the most ch-a, and D(33) showed the most $H_2O_2$. For Gemmeza-11 cultivar, D(22) showed the highest content of carotenoids, MDA and $H_2O_2$, and D(33) resulted in the highest values of ch-a, ch-b and total chlorophyll. Frankly, future open-field trials should be carried out under various abiotic stresses, such as salinity, to assess the effects of plasma on seed robustness and plant growth and productivity.

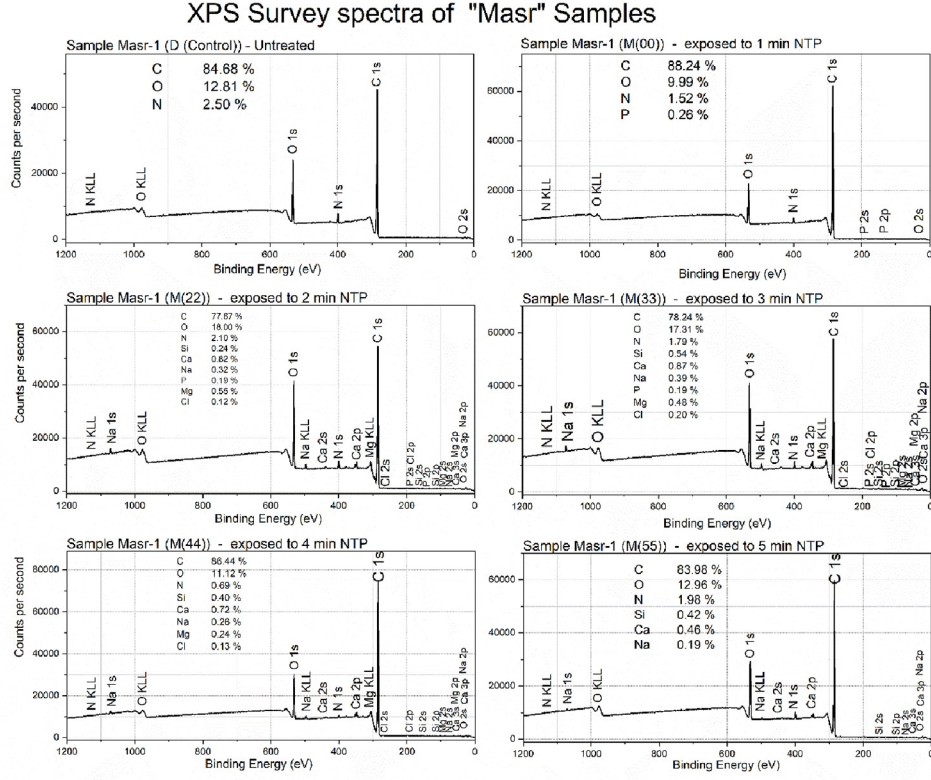

**Fig 7. The XPS survey spectra of treated and control Misr-1 Not Masr samples.** D(00), D(11), D(22), D(33) and D (44) not M(00), M(11), M(22), M(33) and M(44), please add D(44) analysis.

## 4 Discussion

Plasma exposure can change seed-surface properties and stimulate seed germination and seedling growth, induce changes in metabolic plant pathways, modulate enzymatic activities and phytohormones, induce stress resistance, and ultimately influence crop productivity [59–64]. Also, plasma treatment affects the wettability of the coated seed. According to our findings exhibited that seed wettability increased with plasma treatment, especially with time of 1 minute and 4 minutes for Misr-1 and 2 min and 4 minute for Gemmeza-11. Herein, cold plasma modified the seed surface (mainly oxidation due to active particles present in plasma), thus

**Table 2. Physiological traits of wheat cultivars (Misr-1 and Gemmeza-11) produced under different plasma exposures in a greenhouse.**

| Samples | Chlorophyll (mg/g) | | | Carotenoid (mg/g) | MDA (nmol/gFW) | $H_2O_2$ (mM g–1 FW) |
|---|---|---|---|---|---|---|
| | Ch-a | Ch-b | Total | | | |
| Misr-1 cultivar | | | | | | |
| D(00) | 0.71±0.06a | 0.54±0.08a | 1.25±0.02a | 0.28±0.01a | 23.37±3.36b | 0.319±0.04b |
| D(22) | 0.76±0.01a | 0.33±0.01c | 1.09±0.01b | 0.29±0.02a | 24.35±3.31b | 0.385±0.07b |
| D(33) | 0.65±0.11b | 0.42±0.08b | 1.07±0.03b | 0.23±0.06b | 31.32±0.85a | 0.712±0.02a |
| Gemmeza-11 cultivar | | | | | | |
| D(00) | 0.80±0.03b | 0.29±0.01c | 1.09±0.03c | 0.32±0.02a | 26.95±1.67a | 0.510±0.04a |
| D(22) | 0.91±0.01a | 0.50±0.01b | 1.41±0.01b | 0.26±0.01b | 17.97±2.38b | 0.429±0.03b |
| D(33) | 0.93±0.01a | 0.60±0.02a | 1.53±0.02a | 0.25±0.03b | 15.80±1.87b | 0.249±0.01c |

hydrophilicity and water uptake were improved [65,66]. The wettability of seeds varied based on the time of plasma exposure. It has been reported that the wettability and imbibition were found to be directly related to the treatment duration, possibly due to the chemical alternation of lipid layers in the seeds [67]. Furthermore, cold plasma affects seed electron paramagnetic resonance signal [68], seed phytohormones and protein expression in seedlings [69], and content of secondary metabolites [70–72]. It has been reviewed that the main concepts and underlying principles of NTP treatment techniques which clarified the considerable effects of NTP treatments on DNA damage, gene expression, enzymatic activities, morphological and chemical traits, seed germination and plant resistance to stress are considered [40]. Investigations of plasma-induced changes in plant physiological and biochemical processes are likely to reveal new facts of fundamental and applied interest [73]. Accordingly, our findings indicated remarkable responses of wheat seeds and seedlings growth to plasma application. Since glow of cold plasma has ions that cause strong surface etching, plasma treatment of wheat seeds (Misr-1 and Gemmeza-11) resulted in chemical re-structuring of the coated surface. In addition, glow plasma contains a higher concentration of relatively aggressive reactive chemical species of higher energy as well as vacuum ultraviolet radiation, which may explain why variations in the nutrient accumulation on the seed coat surface we observed depended on the plasma treatment. The glow region of plasma can cause significant seeds heating due to strong ion shelling [59,74]. Moreover, plasma treatments give rise to the fixation of atmospheric nitrogen, accumulating nitrite, nitrous acid, nitrate, and nitric acid, nourishing the material [75].

The results indicate also that the response of seeds and seedlings to cold plasma treatment cannot be generalized to all seed species based on the results obtained from only one cultivar. This is because every seed has its own structure, and so exposure to the same conditions will have different effects. Not only genetic potential affects the seed response to plasma but also, the environmental factors such as soil and climatic conditions. Our findings could be the first attempt to prove the relative relation between plasma treatment and plant genotype. It has been clarified that each wheat cultivar has different coping mechanisms and interactions with highly reactive oxygen plasma species. Different responses to plasma could be attributed to the diversified chemical structures of the seed surface, seed size, in addition to genetic makeup [76]. Depression in seedling growth proved that Gemmeza-11 cultivar is more sensitive to the increase in plasma dose. In this context, the seedling length and weight were highly lower after plasma treatment for 3 minutes, associating the reduction in MDA and $H_2O_2$. Misr-1 showed a better responsiveness and adaptability to cold plasma treatment compared to Gemmeza-11. The varietal differences in water and nutrient stress could be attributed to different root system under unfavorable conditions [77]. Therefore, the potentiality of wheat cultivar to tolerate the abiotic stress could be a key factor in seed response to cold plasma treatment [78]. Findings revealed also that seed properties can be optimized by optimizing plasma dose to ensure that the coating treatment matches the needs of a specific cultivar.

On the other hand, a low molecular-mass biopolymer containing lignin-cellulose had a major impact on the wetting-drying cycle of the wheat seeds tested. To enhance seed functionality, modified lignin and lignocellulosic material [79] could be exploited in coating the wheat seed surface with a protective layer prior to NTP treatment. The addition of a low molecular-mass biopolymer containing lignin-cellulose exposed under D(22) after a single wetting-drying cycle increased the water stability more than 200 times. NMR spectroscopy is a powerful analytical tool that has been extensively used to study chemical composition and molecular identity [80,81]. The main advantages of NMR spectroscopy are that it is a nondestructive and highly reproducible method where samples can be studied in both solid and liquid states

[82,83]. The small differences in the peak's intensity at aliphatic and aromatic regions might be associated with the coating layer of lignin on the surface of the treated wheat seeds, which may improve seed resistance to pathogens or drought [84,85]. Wettability is not the only factor that influences seed germination. Others include pH, salt concentration, osmotic potential, C, O and N content, bulk density, porosity, and particle size distribution, and these should be considered in lignin-cellulose biofilm design.

Finally, plasma treatment is known to improve the surface hydrophilicity and water uptake of seeds. These effects are mediated by the functionalization and etching of the seed surface [42]. We observed these effects on the seed surface structure using SEM analysis, which showed that a significant coating of lignin is around 30 μm. For Misr-1, NTP at D(00) and D(22) resulted in tolerance to soil salinity. A thinner thickness required a longer plasma dose (D(33)), but too long a dose (D(44)) resulted in less coating and less tolerance to salinity. For Gemmeza-11, a coating of about 50 μm plasma was effective at D(00) and D(22). Further studies are needed to examine whether different attributes (*i.e.* resistance to bacteria or resistance to drought or salinity) require treatment with different plasma doses.

## 5 Conclusions

The current work presents the first report of combining the use of organic materials extracted from tree waste residues with plasma treatment for seed coating. This seed coating approach offers many advantages for seeds. The coating provides a protective layer that helps adaption in harsh conditions such as high salinity. The organic layer also helps to absorb water and preserve it around the seeds, which helps in germination. Also, this layer forms nutrients for the plant. It acts as a fertilizer for the plant and a food for the bacteria that secrete nutrients for the plants. Thus, the treating seed with both lignocellulose and cold plasma may contribute to the production of seeds robust against undesirable environmental conditions, and open opportunities for the mass cultivation of strategic crops like wheat in stressed environments such as saline soils. Further investigations should be adopted to apply the promising findings of plasma plus lignin coated seeds under open field experimentation.

## Supporting information

**S1 Fig. Solid-state 13C NMR spectra of control (blue) and one-minute treated samples (red).** The figure shows very similar features of both spectra with slightly higher intensity at aliphatic and aromatic regions.
(DOCX)

**S2 Fig. High resolution XPS C 1s spectra of Misr samples.**
(DOCX)

**S3 Fig. High resolution XPS C 1s spectra of Gemmeza-11 samples.**
(DOCX)

## Acknowledgments

The authors thank and appreciate Diener Company's assistance in helping us design a plasma device for treating seeds for use on medium agricultural lands. We urge the company to develop plasma devices that can process large amounts of seeds and are appropriate for cultivation in larger areas.

## Author Contributions

**Conceptualization:** Abd Elfattah T. Elgendy, Hani S. Saudy, Nimer Wehbe, Mohamed Ben Hassine, Ruba Al-Nemi.

**Data curation:** Abd Elfattah T. Elgendy, Hani S. Saudy, Ruba Al-Nemi, Abdul-Hamid Emwas.

**Formal analysis:** Hesham Elsaid, Hani S. Saudy, Mariusz Jaremko.

**Funding acquisition:** Mohamed Ben Hassine, Mariusz Jaremko, Abdul-Hamid Emwas.

**Investigation:** Abd Elfattah T. Elgendy, Hani S. Saudy.

**Methodology:** Hani S. Saudy, Nimer Wehbe.

**Project administration:** Hani S. Saudy.

**Resources:** Abd Elfattah T. Elgendy, Nimer Wehbe, Mohamed Ben Hassine, Ruba Al-Nemi, Mariusz Jaremko, Abdul-Hamid Emwas.

**Software:** Abd Elfattah T. Elgendy, Hesham Elsaid, Mohamed Ben Hassine, Mariusz Jaremko.

**Supervision:** Hani S. Saudy.

**Validation:** Hesham Elsaid, Hani S. Saudy, Abdul-Hamid Emwas.

**Visualization:** Mohamed Ben Hassine, Ruba Al-Nemi.

**Writing – original draft:** Abd Elfattah T. Elgendy, Nimer Wehbe, Ruba Al-Nemi, Mariusz Jaremko, Abdul-Hamid Emwas.

**Writing – review & editing:** Hani S. Saudy, Ruba Al-Nemi.

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
