## [Decision Letter · Decision Letter 0]

5 Jun 2024

PONE-D-24-08803Undergoing Lignin-Coated Seeds to Cold Plasma to Enhance the Growth of Wheat Seedlings and Obtain Future Outcome under Stressed EcosystemsPLOS ONE

Dear Dr. Saudy,

Thank you for submitting your manuscript to PLOS ONE. After careful consideration, we feel that it has merit but does not fully meet PLOS ONE’s publication criteria as it currently stands. Therefore, we invite you to submit a revised version of the manuscript that addresses the points raised during the review process.

The reviewers have recommended revisions to your manuscript. Therefore, I invite you to respond to the reviewers' comments and revise your manuscript.

With Thanks

We look forward to receiving your revised manuscript.

Kind regards,

Diaa Abd El-Moneim

Academic Editor

PLOS ONE

Journal Requirements:

2. Thank you for stating the following financial disclosure: "The research was funded via Science & Technology Development Fund (STDF), Ministry of Scientific Research Egypt, for financial support and providing required research facilities through the approved project, ID: IG-43568." 

Additional Editor Comments:

There are significant concerns about the manuscript's grammar, usage, and overall readability. We, therefore, request that you revise the text to fix the grammatical errors and improve the overall readability of the text.

With Thanks

Reviewers' comments:

Reviewer's Responses to Questions

**Comments to the Author**

1. Is the manuscript technically sound, and do the data support the conclusions?

Reviewer #1: Yes

Reviewer #2: Yes

2. Has the statistical analysis been performed appropriately and rigorously? 

Reviewer #1: No

Reviewer #2: Yes

3. Have the authors made all data underlying the findings in their manuscript fully available?

Reviewer #1: Yes

Reviewer #2: Yes

4. Is the manuscript presented in an intelligible fashion and written in standard English?

Reviewer #1: Yes

Reviewer #2: Yes

5. Review Comments to the Author

Reviewer #1: The research manuscript "Undergoing Lignin-Coated Seeds to Cold Plasma to Enhance the Growth of Wheat Seedlings and Obtain Future Outcome under Stressed Ecosystems" investigates the effects of cold plasma on lignin-coated wheat seeds to enhance growth in stressed ecosystems. The study presents a novel and innovative method in addressing the challenges of seed germination in stressed ecosystems by introducing cold plasma as a treatment to improve seed vigor.

The physiological analyses presented in the paper provide a comprehensive understanding of the effects of cold plasma treatment and lignin coating on seed germination and growth. The study evaluated various parameters, including seed germination percentage, growth rate, and antioxidant activity to assess seed vigor following the cold plasma and lignin coating treatments. These analyses provide significant insights into the mechanisms involved in the enhancement of wheat seed growth in stressed ecosystems.

Regarding the seed germination experiment, the manuscript does not specify the seed germination protocol utilized. Therefore, it is unclear if ISTA or AOSA guidelines were followed, which might lead to concerns about replicability and comparability of the results. The manuscript should clarify the applied protocol and provide specific details for replicability purposes.

The sample size used for the vigor assessment is limited to five seeds per treatment. While this is an acceptable number for preliminary screening of germination and growth potential, it is considered small for solid scientific conclusions. The manuscript should justify the reasoning behind the use of such a small sample size and provide further analyses, including standard deviation and statistical significance to support the presented data.

The manuscript does not explicitly identify prominent environmental stressors considered during the study, which limits the applicability and generalizability of the obtained results. This makes it challenging to infer the relevance of the study's findings to actual ecological systems. The manuscript should specify the significant environmental stressors considered during the study and contextualize the results with their relevance to the relevant ecosystem.

In conclusion, the "Cold Plasma Treatment of Lignin-Coated Wheat Seeds Enhances Growth in Stressed Ecosystems" study presents a novel and innovative approach to improve seed germination and growth in stressed ecosystems. The study's physiological analyses provide useful insights into the effects of cold plasma treatment and lignin coating on seed vigor. While the study has some limitations, including concerns about the sample size and lack of clarity regarding the used seed germination protocol and environmental stressors, the research presents a crucial contribution to the field of seed germination under challenging ecological conditions.

Reviewer #2: The discussion section is weak. There is no attempt to establish a connection between the literature and the findings. A few pieces of literature are mentioned first, followed by the findings, but similarities and differences are not discussed. The discussion section, which is the most important part and can contribute the most to science, prompting researchers to think, innovate, and generate new ideas, has been kept very brief in this publication. Although this was a study done with effort, it requires major revision. There is no real discussion in the discussion section. I do not believe that so many people have read this paper. At least one of them should know that this paper, in its current form, should not be submitted to the PLOS ONE journal. It is likely that a very junior academic wrote this paper and added the names of other colleagues. It is not recommended for publication in its current form. I request that it be sent back to me after major revision.

6. PLOS authors have the option to publish the peer review history of their article (what does this mean?). If published, this will include your full peer review and any attached files.

Reviewer #1: No

Reviewer #2: **Yes: **sadettin

---

## [Author Response · Author response to Decision Letter 0]

23 Jun 2024

We addressed all required corrections and responded positively to all suggestions of reviewers

---

## [Decision Letter · Decision Letter 1]

22 Jul 2024

Undergoing Lignin-Coated Seeds to Cold Plasma to Enhance the Growth of Wheat Seedlings and Obtain Future Outcome under Stressed Ecosystems

PONE-D-24-08803R1

Dear Dr. Saudy,

We’re pleased to inform you that your manuscript has been judged scientifically suitable for publication and will be formally accepted for publication once it meets all outstanding technical requirements.

Kind regards,

Diaa Abd El-Moneim

Academic Editor

PLOS ONE

Additional Editor Comments (optional):

Reviewers' comments:

Reviewer's Responses to Questions

**Comments to the Author**

1. If the authors have adequately addressed your comments raised in a previous round of review and you feel that this manuscript is now acceptable for publication, you may indicate that here to bypass the “Comments to the Author” section, enter your conflict of interest statement in the “Confidential to Editor” section, and submit your "Accept" recommendation.

Reviewer #1: All comments have been addressed

Reviewer #2: All comments have been addressed

2. Is the manuscript technically sound, and do the data support the conclusions?

Reviewer #1: Yes

Reviewer #2: Yes

3. Has the statistical analysis been performed appropriately and rigorously? 

Reviewer #1: Yes

Reviewer #2: Yes

4. Have the authors made all data underlying the findings in their manuscript fully available?

Reviewer #1: No

Reviewer #2: Yes

5. Is the manuscript presented in an intelligible fashion and written in standard English?

Reviewer #1: Yes

Reviewer #2: Yes

6. Review Comments to the Author

Reviewer #1: (No Response)

Reviewer #2: The athours did entire correction. this is the second round, the authors did the corrections so no need to say more

7. PLOS authors have the option to publish the peer review history of their article (what does this mean?). If published, this will include your full peer review and any attached files.

Reviewer #1: No

Reviewer #2: No

---

## [Editor Report · Acceptance letter]

24 Jul 2024

PONE-D-24-08803R1 

PLOS ONE

Dear Dr. Saudy, 

I'm pleased to inform you that your manuscript has been deemed suitable for publication in PLOS ONE. Congratulations! Your manuscript is now being handed over to our production team.

Kind regards, 

on behalf of

Dr. Diaa Abd El-Moneim 

Academic Editor

PLOS ONE